# *WRKY22* Transcription Factor from *Iris laevigata* Regulates Flowering Time and Resistance to Salt and Drought

**DOI:** 10.3390/plants13091191

**Published:** 2024-04-25

**Authors:** Lijuan Fan, Zhaoqian Niu, Gongfa Shi, Ziyi Song, Qianqian Yang, Sheng Zhou, Ling Wang

**Affiliations:** College of Landscape Architecture, Northeast Forestry University, Harbin 150040, China; flj@nefu.edu.cn (L.F.); niuzhaoqian@nefu.edu.cn (Z.N.); sumu@nefu.edu.cn (G.S.); 2000lhszy@nefu.edu.cn (Z.S.); 16650601598@nefu.edu.cn (Q.Y.); liuguiling90@126.com (S.Z.)

**Keywords:** *Iris*, *WRKY* transcription factor, flowering time regulation, salt stress, drought stress

## Abstract

*Iris laevigata* Fisch. is an excellent ornamental plant in cold regions due to its unique ornamental ability and strong cold resistance. However, the flowering period of the population is only about 20 days, greatly limiting its potential uses in landscaping and the cutting flower industry. In addition, *I. laevigata* is often challenged with various abiotic stresses including high salinity and drought in its native habitats. Thus, breeding novel cultivars with delayed flowering time and higher resistance to abiotic stress is of high importance. In this study, we utilized sequencing data from the *I. laevigata* transcriptome to identify WRKYs and characterized *IlWRKY22*, a key transcription factor that modulates flowering time and abiotic stress responses. *IlWRKY22* is induced by salt and drought stress. We cloned *IlWRKY22* and found that it is a Group IIe *WRKY* localized in the nucleus. Overexpressing *IlWRKY22* in *Arabidopsis thaliana* (L.) Heynh. and *Nicotiana tabacum* L. resulted in a delayed flowering time in the transgenic plants. We created transgenic *N. tabacum* overexpressing *IlWRKY22*, which showed significantly improved resistance to both salt and drought compared to the control plants. Thus, our study revealed a unique dual function of *IlWRKY22*, an excellent candidate gene for breeding novel *Iris* cultivars of desirable traits.

## 1. Introduction

A key event in the life cycle of angiosperms is the formation of flowers, which marks the transitions of plants from vegetative growth to reproductive growth [1]. It not only directly affects the growth, development, and yield of crop plants but also has a great impact on horticultural plants as the flower is the most important ornamental part in landscaping [2]. Flowering is controlled by a very large and fine-tuned regulatory network, which integrates various exogenous and endogenous factors to determine the most appropriate flowering time [3,4,5]. In response to various environmental factors that influence flowering, plants have developed multiple pathways with multiple transcription factors (TFs) for the transition to flowering. The critical roles of TFs in flowering have been extensively studied [6]. Some of the most important proteins in flowering pathways are TFs or transcriptional regulators including constans (CO) in the photoperiod pathway and flowering locus C (FLC), a MADS-box TFs that represses flowering. Major flowering pathways include the photoperiod pathway, which involves the de-repression of cycling of factors (CDF) on CO and flowering locus T (FT), as well as the vernalization, temperature, and autonomous pathways, which involve the de-repression of FLC on FT. Additionally, the GA pathway involves the de-repression of suppressor of overexpression of constans1 (SOC1) through the degradation of DELLA proteins [7].

On the other hand, various environmental stresses such as high temperature, cold, salinity, and drought can inhibit the metabolism and growth in plants [8]. High salinity and drought are two main abiotic stresses that affect plant growth and development and even cause death [9]. Salt stress causes ion imbalance, osmotic stress, and secondary oxidative stress, resulting in impaired photosynthetic capacity and intracellular nutrient metabolism imbalance and ultimately affecting plant development and yield. Under high salinity, ions from the soil accumulate in the cytoplasm and thus disrupt the osmotic regulation. Ion imbalance can last a few days or even weeks in tender parts of the plants such as the bud [10]. It further induces secondary oxidative stress, leading to the inhibition of photosynthesis and metabolism and ultimately a deduction in plant yield [11]. Similar to high salinity, drought causes osmotic stress and secondary stress including oxidative stress, damage to cellular components, and metabolic dysfunction [10]. Therefore, drought and salt stress have certain overlapping signals [12]. In addition, evolution and adaptation equip plants to quickly sense stress signals via TFs such as WRKYs and gene regulatory networks [13,14]. *WRKY* exhibits a rapid response to adverse environments and plays a crucial role in enhancing the tolerance of various plants to abiotic stress [4]. TFs play a critical role in the responses to abiotic stresses by activating or inhibiting the expression of target genes, leading to reprogramming in plant metabolism and adaptability to abiotic stress [15]. Thus, there is an increased interest in using TFs to combat environmental stresses in the face of global warming [16,17]. For instance, *HaWRKY22* enhances the drought stress tolerance of *Helianthus annuu* L. [18], while *PsWRKY22* from *Prunus salicina* Lindl. enhances tolerance to both salt and drought stress [19].

Thus, the two seemingly unrelated processes of flowering and response to salt and natural drought stress have one thing in common: both are under tight control of TFs. Interestingly, TFs may regulate multiple independent biological processes because they can interact with distinct sets of substrates. The WRKY family is a good example that controls a diverse range of physiological events via a complex gene network [20,21]. For instance, *WRKY* regulates plant tolerance to various stresses such as low temperature, salt, high temperature, drought, and flowering by interacting with abscisic acid (ABA), jasmonic acid (JA), and ROS. WRKYs are named after the highly conserved WRKYGQK domain at the N-terminus [22]. The C terminus contains a CX4-7CX22 or 23HXH/C zinc finger that is important for protein interactions and DNA binding [20,23]. Based on the number of WRKY domains and the type of zinc finger structure, WRKYs are categorized into three groups: group I (two WRKY domains and a C2H2-type zinc finger structure), group II (one WRKY domain and a C2H2-type zinc finger structure), and group III (one WRKY domain and a C2HC-type zinc finger structure) [24]. With the advancement of sequencing technologies, WRKYs have been identified on a genome-wide scale in many plant species. For example, there are 74 members in *A. thaliana*, 59 in *Vitis vinifera* L., 127 in *Malus pumila* Mill., 102 in *Oryza sativa* L., and 119 in *Zea mays* L. [25].

*I. laevigata* is native to northeast China with strong cold resistance and beautiful flowers of unique shape and bright color [26]. Such desirable traits make *I. laevigata* a good candidate for both landscaping and industrial use as a cutting flower. However, the potential is dampened because of the short flowering period with only 2 days for a single flower and 20 days for the population [26]. To stagger the flowering time and prolong the flowering period, it is necessary to find genes that can advance or delay the flowering. *I. laevigata* is also sensitive to abiotic stress [27]. Thus, extending the flowering period and increasing the resistance to salt and drought stress are two central goals in the field of breeding *I. laevigata* of novel properties [28,29]. One prerequisite to achieve these goals is to identify key genes controlling the target traits, which has been propelled tremendously by the advances in transcriptome sequencing [30,31,32].

However, molecular studies in *I. laevigata* are difficult due to the lack of genome information [26]. Our group pioneered the first transcriptomics studies in the tepals of *I. laevigata*. In this study, we report the identification of 68 WRKYs in *I. laevigata*, which are distributed to the three groups that regulate multiple biological processes. Among them, group II genes primarily participate in defense responses against bacterial and fungal infections, abiotic stress reactions, aging processes, and developmental phenomena [33]. *WRKY22*, a member of the group II transcription factor family within the WRKY family, has been demonstrated to play a pivotal role in conferring resistance against bacteria [34], low temperature [35], and abiotic stressors [18,19], as well as governing the regulation of flowering [36]. To promote resistance and flowering regulation of swallow flowers, this study focused on the functional validation of *IlWRKY22*.

We found *IlWRKY22* was significantly up-regulated upon salt and drought stress treatment. Furthermore, we cloned *IlWRKY22* and overexpressed it in *N. tabacum*, resulting in enhanced salt and drought tolerance. Overexpressing *IlWRKY22* in *A. thaliana* and *N. tabacum* led to a delay in flowering, which was accompanied by the dysregulation of many flowering transcription factors including CO and GA-20 oxidase (GA20OX), SOC1, and squamosa promoter binding protein like3 (SPL3). Thus, our data demonstrated that *IlWRKY22* is a negative regulator in flowering and a positive regulator in resistance to salt and drought. In conclusion, *IlWRKY22* is a multifaceted TF that can be tweaked to improve multiple traits. Our findings have the potential to extend the flowering period of *I. laevigata* through the cultivation of late-flowering transgenic plants alongside wild type (WT) plants. Because the transgenic plants have stronger salt tolerance and drought resistance, the overall adaptability of *I. laevigata* will also be improved. Together, our results hold promise for the wider implementation of *I. laevigata* in landscaping projects throughout northeastern China.

## 2. Materials and Methods

### 2.1. Plant Materials

*I. laevigata* was outdoor-planted in the nursery of the School of Landscape Architecture, Northeast Forestry University (126°64′ E, 45°72′ N, Heilongjiang Province, China).

Seeds of *A. thaliana* (ecotype Columbia) Col-0 were surface-sterilized by immersing them in 75% alcohol (Lircon, Dezhou, China) for 1 min, followed by triple rinsing with sterile water. Seeds were then disinfected in 0.8% NaClO (Xilong Scientific, Beijing, China) for 10 min and rinsed 5 times again. Sterilized seeds were sowed on 1/2 MS (Hope Bio-Technology, Qingdao, China), and vernalized at 4 °C in the dark for 2 days, followed by cultivation at a greenhouse under 16 h/8 h (light/dark) at 20 °C. After 7–9 days, seedlings were transferred into a potting mix (peat soil/vermiculite/perlite of 5/3/2) and allowed to grow for 3 weeks before transformation.

*N. tabacum* seeds were disinfected with 75% alcohol (Lircon, Dezhou, China) for 1 min, followed by triple rinsing with sterile water. Seeds were then disinfected with 2% NaClO (Xilong Scientific, Beijing, China) for 10 min and rinsed with sterile water 5 times. Seeds were cultured in MS (Hope Bio-Technology, Qingdao, China) at 25 °C, with 14 h/10 h (light/dark). Plants of 4–6 true leaves were used for transformation.

### 2.2. The Treatment of I. laevigata under Salt and Drought Stress

In order to further investigate the involvement of *IlWRKY22* in the responses of *I. laevigata* to salt and drought stresses, 1-year-old *I. laevigata* plants with carefully selected for a 7-day water culture experiment. For the salt treatment, the plants were exposed to a concentration of 300 mM NaCl, while for the drought treatment, they were subjected to a concentration of 20% PEG-6000. The expression level of *IlWRKY22* in the leaves was then measured in real-time at 0, 2, 6, 12, and 24 h.

### 2.3. Gene Sequence Identification and Phylogenetic Analysis

Putative *WRKY* sequences were identified from the transcriptome data from the tepals of *I. laevigata* with the online annotation tool from the Beijing Genomics Institute (Beijing, China) (https://www.bgi.com/home, accessed on 3 January 2023). We downloaded all *A. thaliana WRKY* sequences from the National Center for Biotechnology Information (NCBI) (https://www.ncbi.nlm.nih.gov/, accessed on 3 January 2023). Phylogenetic analysis was conducted in MEGA 5.0 (Version 10.1.8) (https://phytozome-next.jgi.doe.gov/, accessed on 3 January 2023) using *WRKY* sequences from both *I. laevigata* and *A. thaliana*. The open reading frames of these sequences were determined by the ORF Finder tool (https://www.ncbi.nlm.nih.gov/orffinder/, accessed on 3 January 2023), and the conserved domains were identified by the CD Search tool (https://www.ncbi.nlm.nih.gov/Structure/cdd/wrpsb.cgi, accessed on 3 January 2023). *WRKY* sequences of different species were obtained by BLAST (https://blast.ncbi.nlm.nih.gov/Blast.cgi, accessed on 3 January 2023) at NCBI. Multiple sequence alignment was performed using DNAMAN (Version 5.2.2) (https://www.lynnon.com/dnaman.html, accessed on 3 January 2023). Sequence alignment was performed using Clustal W (http://www.clustal.org/clustal2/, accessed on 3 January 2023), followed by the construction of a phylogenetic tree using the neighbor-joining (NJ) method with 1000 bootstraps. The resulting tree was then annotated using iTOL (https://itol.embl.de/, accessed on 3 January 2023).

### 2.4. Gene Cloning and Sequence Analysis

Total RNA was extracted from the tepals of *I. laevigata* using a kit (Kangweishiji, Taizhou, China), and the integrity was checked by electrophoresis. Frist-strand cDNA was synthesized using the Primer Script TM RT kit (TakaRa, Beijing, China). *IlWRKY22* (accession number: ON399552) was cloned using KOD-plus-neo (ToYoBo, Shanghai, China) with primers (Ruibo Kexing, Harbin, China) listed in Appendix A. The cloned sequences were inserted into the Cloning vector pEASY^®^-Blunt Zero Cloning Kit (TransGen, Beijing, China) and transformed into *Escherichia coli* DH5α (WEIDI, Shanghai, China). The pCAMBIA1300: *IlWRKY22-GFP* carrier was constructed for heterologous expression through homologous recombination. Homologous recombination primers (Ruibo Kexing, Harbin, China) listed in Appendix A. The pCAMBIA1300 vector were digested with SalI and BamHI, which were then used to create pCAMBIA1300-*IlWRKY22-GFP* using ClonExpressII (Vazyme, Nanjing, China). The recombinant vector was transformed into *E. coli* DH5α (WEIDI, Shanghai, China) for sequencing. Positive plasmids were transferred into *Agrobacterium tumefaciens* GV3101 (WEIDI, Shanghai, China) for subsequent infection experiments. Bioinformatics analysis of *IlWRKY22* was performed as described [26].

### 2.5. Subcellular Localization

The online tool Cell-PLoc 2.0 was used to predict subcellular localization (http://www.csbio.sjtu.edu.cn/bioinf/Cell-PLoc-2/, accessed on 3 January 2023). The subcellular localization of WRKY22 was experimentally determined as described [37]. Briefly, bacteria GV3101 (WEIDI, Shanghai, China) harboring pCAMBIA1300: *IlWRKY22-GFP* were cultured to an OD_600_ of 0.6–0.8 and collected. After resuspending in 10 mM MgCl_2_ to an OD_600_ of 1.5, acetosyringone (Sinopharm, Beijing, China) was added to a final concentration of 200 µM. The bacterial solution was activated for 3 h in the dark and then injected to the lower side of *Nicotiana benthamiana* Domin. leaves using a syringe. Expression of the fusion protein WRKY22-GFP was visualized under an AN-28L fluorescent microscope (Aonuoguangxu, Shenzhen, China) in the transformed epidermal cells [26].

### 2.6. Plant Transformation

Transgenic *A. thaliana* plants were created using the flower dip method reported before [38]. Seeds of transformed plants were selected on 1/2 MS medium supplemented with 25 mg/L Hyg (Xilong Scientific, Beijing, China). After 7–9 days, T1 seedlings that survived on the screening medium were transferred to a potting mix of peat soil, vermiculite, and perlite in a ratio of 3:2:1. The plants were then cultured at a temperature of 20 °C with a photoperiod of 16 h/8 h (light/dark). Transgenic plants were then screened by PCR, the primers (Ruibo Kexing, Harbin, China) listed in Appendix A. Homozygous lines were obtained by 3 consecutive generations of screening. Three lines exhibiting high expression levels were selected.

Agrobacterium-mediated transformation of *N. tabacum* leaf disk and screening of the resulting transgenic plants were performed according to a previously published method [39].

### 2.7. Determination of Flowering Phenotypes and Gene Expression

Bolting time and flowering time were recorded of each plant, which were observed and counted daily. The survey criteria for bolting time was the time required from seeding to bud visibility. Flowering time was measured as the time required from sowing to the full opening of the first flower [40,41,42,43,44]. The number of rosette leaves was also recorded. The expression levels of CO, GA20OX, vernalization1 (VRN1), SPL3, flowering control local A (FCA), short vegetative phase (SVP), trehalose-6-phosphate synthase1 (TPS1), SOC1, and FLC in *A. thaliana* seedlings at 7, 10, and 14 days of age were determined using real-time quantitative PCR (RT-qPCR) with the 2^−ΔΔCT^ method; the primers (Ruibo Kexing, Harbin, China) are listed in Appendix A.

### 2.8. The Treatment of Transgenic N. tabacum Salt and Drought Stress

To ensure the accuracy of the data, we set up 3 repeat groups for each line, with 5 plants in each group. Salt and drought stress tests were performed on *IlWRKY22* overexpressed *N. tabacum* using EV and WT controls. Initially, the plants were cultivated in a potting mix consisting of peat soil, perlite, and vermiculite (3:1:1), and under a photoperiod of 16 h of light followed by 8 h of darkness at 28 °C, for a duration of 30 days. Subsequently, plants with comparable overall health were subjected to salt and drought stress treatments. For the salt treatment, plants were watered with 100 mL of a 300 mM NaCl solution every 2 days. In the case of natural drought, plants were deprived of water. Phenotypic observations and leaf samples were collected at 0, 7, and 14 days after the initiation of the treatments.

### 2.9. Photosynthetic Parameters in N. tabacum under Salt and Drought Stress

A Li-6400 (ecotek, Beijing, China) portable photosynthesis instrument was utilized to quantify various photosynthetic parameters including net photosynthetic rate (Pn), stomatal conductance (Gs), intercellular CO_2_ concentration (Ci), and transpiration rate (Tr) from 9:00 am to 10:00 am. Three individual plants were randomly selected from each line (OE, EV, and WT), and the third and fourth leaf were used for measurement. The Li-6400 instrument was calibrated before taking measurements, and data was recorded after the reading was stable. Five measurements were taken for each leaf [39]. The chlorophyll (Chl) content was assessed using the acetone extraction method [45,46]. The functional leaves of the same part were selected, and the chlorophyll fluorescence parameters of the leaves were determined by using a PAM-2500 (WALZ, Effeltrich, Germany) portable modulated chlorophyll fluorometer [39].

### 2.10. Other Physiological Measurements under Salt and Drought Stress N. tabacum Plants

Commercially available kits (catalog number G0105F, and the same hereafter) from Grace Biotechnology (Shanghai, China) were used for the quantification of CAT, hydrogen peroxide (G0112F), and $O_{2}^{-}$ (G0116F). The determination method referred to the instructions of the kit. Malondialdehyde (MDA), superoxide dismutase (SOD) activity, and peroxidase (POD) activity were measured based on a previous study [39]. To prepare the stressed *N. tabacum* leaves, leaf discs with a diameter of 1.5 cm were cut, following the protocol described by Wang [39]; p-Nitro-Blue tetrazolium chloride (NBT) and 3,3′-diaminobenzidine tetrahydrochloride (DAB) staining techniques were employed to assess the concentration of superoxide anion and hydrogen peroxide in the leaves, respectively.

### 2.11. Reverse Transcription PCR (RT-PCR) and RT-qPCR

We set up 3 repeat groups for each line, with 5 plants in each group. The expression patterns of *IlWRKY22* were measured in various parts of *I. laevigata*, including roots, tubers, and leaves under normal growth conditions. Additionally, the response of *IlWRKY22* in leaves of *I. laevigata* 0, 2, 6, 12, and 24 h after salt and drought stress were measured. Randomly selected T3 lines of *A. thaliana* and *N. tabacum* were used for measuring the expression of *IlWRKY22*. Furthermore, the expression levels of stress response genes (*NtCAT*, *NtHAK1*, *NtPMA4*, *NtPOD*, *NtSOD*, and *NtSOS1*) were measured in *N. tabacum* leaves under NaCl and drought stress for 0, 7, and 14 days. This analysis was conducted in WT, EV, and 3 overexpressed *IlWRKY22* lines. The method used for total RNA extraction and reverse transcription to cDNA was the same as described above. *IlPP2A*, *ACT2*, and *NtTUBA* served as internal reference genes for *I. laevigata*, *A. thaliana*, and *N. tabacum*, respectively. RT-qPCR was performed using the SYBR Green I method, following the instructions provided by UltraSYBR Mixture (CWBIO). The results were calculated as 2^−ΔΔCT^; the primers (Ruibo Kexing, Harbin, China) are listed in Appendix A.

### 2.12. Statistical Analysis

Data recording and statistical analysis were conducted using Excel (Version 2021). Statistical analyses were performed using SPSS (version 26). Before performing the analysis of variance, the experimental data were subjected to the Shapiro–Wilk and Levene tests to verify residual normality and homoscedasticity, respectively. We used one-way ANOVA of statisical analysis and Duncan’s test. The data were tested for statistical significance within a confidence interval using the least significant difference method (*p* < 0.05). Plots were generated using Origin (version 2021). Statistical differences between the transgenic and control plants were determined.

## 3. Results

### 3.1. Identification of WRKY Genes in I. laevigata

We performed transcriptome data mining of *WRKY* sequences in the tepals of *I. laevigata* (unpublished). WRKY family were obtained by BLAST and sequence analysis. We found a total of 68 putative members (Figure 1). Phylogenetic analysis of the coded proteins showed that they are distributed to groups I (n = 22), IIa (n = 5), IIb (n = 7), IIc (n = 1), IId (n = 13), IIe (n = 9), and III (n = 11). By contrast, the 59 *WRKY* members in *A. thaliana* showed a different pattern of distribution among the groups: groups I (n = 11), IIa (n = 2), IIb (n = 7), IIc (n = 16), IId (n = 6), IIe (n = 6), and III (n = 11). Given the distinctive roles of each group, this may indicate functional divergence of *WRKY* genes in *I. laevigata.*

### 3.2. Cloning of IlWRKY22 and Subcellular Localization

We next cloned *WRKY22* using *I. laevigata* cDNA with PCR. The sequence of *IlWRKY22* was 840 bp. Sequence alignment showed that it contains the highly conserved WRKYGQK domain (Figure 2A) and it is most similar to *WRKY22* from *Zingiber officinale* (Figure 2B) with the C2H2 zinc finger [20], a typical feature of the group IIe of the *WRKY* family.

The IlWRKY22 protein was predicted to be in the nucleus. To verify this in vivo, we transiently expressed an IlWRKY22-GFP fusion protein in *N. benthamiana* leaves. We found the fusion protein is exclusively expressed in the nucleus of epidermal cells (Figure 2C). Thus, this data supported the prediction and agreed with the putative TF function of IlWRKY22.

### 3.3. Expression of IlWRKY22 in I. laevigata

We treated one-year-old *I. laevigata* with NaCl at 300 mM and PEG-6000 at 20%. The expression of *IlWRKY22* was examined in the roots, tubers, and leaves in *I. laevigata*. A higher level of *IlWRKY22* was found in the leaves (Figure 3A). The real-time expression level of *IlWRKY22* in the leaves was detected after 0, 2, 6, 12, and 24 h; the peak expression was found 6 h after treatment with salt and 2 h after drought treatment (Figure 3B,C). Five plants were selected from each line of each treatment gradient for measurements, which were repeated three times.

### 3.4. Overexpressing IlWRKY22 in Both A. thaliana and N. tabacum

We obtained 14 positive transgenic *A. thaliana* lines and 10 *N. tabacum* lines, respectively, via Hyg (Xilong Scientific, Beijing, China) selection and PCR verification. These plants were allowed to self-cross for two consecutive generations, and randomly selected T3 lines of *A. thaliana* and *N. tabacum* were used for further experiments. RT-qPCR showed much-enhanced expression levels of *WRKY22* in these transgenic lines compared to the WT and empty vector (EV); three lines of both *A. thaliana* (designated as OE-1, OE-4, and OE-9) and *N. tabacum* (designated as OE-2, OE-3, and OE-5) (Appendix A) with high expression level were selected for the follow-up experiment.

### 3.5. Overexpressing IlWRKY22 Delays Flowering in A. thaliana and N. tabacum

We used two plant model systems for functional validation studies. For *A. thaliana*, we focused more on the flowering phenotype. The use of *N*. *tabacum* for stress treatment is convenient for subsequent measurement of multiple indicators; we also used *N*. *tabacum* to monitor the flowering phenotype. Firstly, we compared the flowering statistics of WT, EV, and *IlWRKY22* overexpressing *A. thaliana* under short-day conditions (Figure 4A). In the WT lines, bolting time was 31.0 d, flowering time was 34.9 d, and the number of rosettes was 14.3. Similarly, the results of EV lines were 31.6 d for bolting time, 35.1 d for flowering time, and 14.8 rosettes (*p* > 0.05). In contrast, overexpression of *IlWRKY22* resulted in a significant (*p* < 0.05) delay in bolting time to 35.8 d, flowering time to 39.1 d, and an increase in the number of roseate leaves to 18.1 (Figure 4B–D). Then we compared the flowering statistics between WT and EV *N. tabacum* as well as *IlWRKY22*-overexpressing *N. tabacum* under long-day conditions (Appendix A). The flowering times observed were found to be not significantly different with values of 62.7 d, 63.4 d for WT and EV lines, while overexpression of *IlWRKY22* further delayed flowering time to reach a value of 68.6 d (Appendix A). In summary, *IlWRKY22* was found to significantly delay flowering.

### 3.6. Overexpression of IlWRKY22 Modulates the Expression of Flowering Time Genes

To understand the molecular mechanisms by which *IlWRKY22* delays flowering, we quantified the mRNA levels of key genes in flowering pathways. On day 7, the RT-qPCR results revealed significant differences in the expression levels of key genes *CO*, *GA20OX*, *SVP*, *FLC*, and *VRN1* within the photoperiod pathway, GA pathway, and temperature pathway among WT, EV, and OE lines (Figure 5A). *CO* and *GA20OX* are significantly up-regulated, while *SVP*, *FLC*, and *VRN1* are significantly down-regulated. This suggested that *IlWRKY22* may regulate these pathways in the early flowering transition. The FLC protein acts as a crucial regulator of flowering, capable of integrating signals from SVP and VRN1 to suppress the expression of *FT*. On day 10, *CO*, *FLC*, *SVP*, *VRN1,* and *GA20OX* are significantly up-regulated, while *FCA* are significantly down-regulated. It is speculated that with the growth of *A*. *thaliana*, the internal transcription environment of the expressing plant is different at different stages, so the expression level changes at day 10. The OE lines exhibited significantly elevated levels of *FLC* expression compared to the control lines (Figure 5B), which was consistent with their observed delayed flowering phenotype. Among all the genes examined, only *GA20OX* was up-regulated in the OE plants (1.4-fold higher) compared to the WT and EV plants at day 14, while all other genes showed lower expression levels in the OE plants (Figure 5C). Notably, *SPL3* showed 80% down-regulation in the OE plants. This indicated that the OE plants were still at the early stage of flowering transition on day 14. Compared to the control group, at day 14, apart from a significant up-regulation of *GA20OX* and no significant difference in *CO* expression levels, all other genes were down-regulated in the OE strain (*p* < 0.05) (Figure 5C).

### 3.7. Overexpressing IlWRKY22 in N. tabacum Enhances Resistance to Salt and Drought Stress

*IlWRKY22* is a multifaceted gene during distinct biological processes. Thus, we next explored its role in abiotic stress [47]. WT and EV were employed as controls; OE-2, OE-3, and OE-5 (Transgenic *N. tabacum* lines with *IlWRKY22* gene) were subjected to salt and drought stress. Five plants were selected from each line of each treatment gradient for measurements, which were repeated three times. All five lines showed similar overall phenotypes under normal conditions (Figure 6A). Under salt stress, after 7 days post-treatment (dpt), the leaf growth rate of WT and EV lines was lower than that of the transgenic lines. At 14 dpt, transgenic lines were significantly taller than the control plants (Appendix A), which showed signs of yellowing and wilting of leaves; however, this led to a more severe inhibition of growth in control plants compared to the transgenic plants. Under natural drought stress, WT and EV lines showed wilting at 7 dpt. All five lines showed wilting at 14 dpt; WT and EV lines showed obvious wilting and significantly lower plant height compared to the transgenic lines.

### 3.8. Impact of Overexpressing IlWRKY22 on Photosynthesis under Salt and Drought Stress

H_2_O and CO_2_ are the basic raw materials for photosynthesis, which sustains normal plant growth. Gs, Chl content, Fv/Fm, Pn, Ci, and Tr are commonly used to reflect the photosynthetic capacity of plants. At the onset of salt treatment, there were no significant differences observed in various indicators (*p* > 0.05). Under salt stress, the Chl content of the five lines showed an increasing trend (Figure 7B), while Pn, Fv/Fm, Gs, Ci, and Tr gradually decreased. With the increase of the stress treatment time, the difference between the longitudinal values of the data on the 0 dpt, 7 dpt, and 14 dpt was more significant, indicating that the impact of salt stress on the photosynthesis of the plant was increasing, seriously limiting the photosynthetic capacity of the plant. However, at 7 and 14 dpt, the transgenic lines exhibited significantly higher Pn, Chl content, and Fv/Fm compared to the control lines. Conversely, Gs, Ci, and Tr were significantly lower in the transgenic lines than in the control lines during these time points (Figure 7A–F). We speculated that this might be a transgenic line that under salt stress, *IlWRKY22* alleviated the decrease in photosynthetic efficiency by increasing or maintaining the Chl content and increased the plants’ water resistance under osmotic stress by closing the stomata to retain water. Similarly, there was no significant difference in the six indexes of the five lines on 0 dpt of drought stress (*p* > 0.05), and the trend was the same as that of salt stress with the increase of drought stress time, which may be because drought stress can trigger ABA synthesis in vascular tissues and guard cells. ABA signaling in guard cells regulates plasma membrane ion channels, triggering long-term outflow of negative ions and K^+^, leading to guard cell shrinkage and stomatal closure [48]. We observed a gradual decline in Pn, Fv/Fm, Gs, Ci, and Tr for all tested lines as stress progressed; these indicators reached very low levels at 14 dpt, when the five lines were already severely dehydrated compared to 0 dpt. Notably, at 7 and 14 dpt, the three overexpressing lines demonstrated significantly higher Pn values along with increased Chl content and Fv/Fm when compared to the control group (*p* < 0.05). Conversely, Gs, Ci, and Tr showed a significant decrease in these overexpressing lines relative to controls under drought conditions (*p* < 0.05) (Figure 7G–L). In summary, *IlWRKY22* can alleviate damage of transgenic *N. tabacum* induced by salt stress and drought stress, which is consistent with the observed phenotype results of transgenic tobacco under stress conditions.

### 3.9. IlWRKY22 Promotes the Response of N. tabacum to Salt and Drought Stress

Salt and drought stress can lead to the production of reactive oxygen species (ROS) in plants, which oxidize lipids and generate MDA. Superoxide anions ($O_{2}^{-}$) and H_2_O_2_ are the main ROS substances [49]. On the other hand, ROS and their derivatives can be counterbalanced by the antioxidant system including SOD, CAT, and POD [50]. To study the involvement of *IlWRKY22* in the ROS clearance enhanced salt and drought stress resistance, we quantified the levels of MDA, $O_{2}^{-}$, and H_2_O_2_, as well the enzymatic activity of SOD, CAT, and POD in the leaf. The results showed that there was no significant difference in MDA, $O_{2}^{-}$, and H_2_O_2_ contents among the five strains under 0 dpt stress. We found increasing levels of MDA, $O_{2}^{-}$, and H_2_O_2_ in all plants as the salt/drought stress progressed (Figure 8A–C,G–I), indicating abiotic stress induced oxidative stress in plants. Under both conditions, the OE plants showed significantly lower levels of MDA, $O_{2}^{-}$, and H_2_O_2_ at both 7 and 14 dpt compared to the control plants (*p* < 0.05). NBT and DAB can directly show the amount of $O_{2}^{-}$ and H_2_O_2_ accumulation through the change of color. At 0 dpt, MDA, $O_{2}^{-}$, and H_2_O_2_ contents of the five lines were not significantly different (*p* > 0.05), and the color of the plants was gradually deepened with the stress time, which indicated that $O_{2}^{-}$ and H_2_O_2_ accumulated more with the passing of time and the plants were subjected to more stress. NBT and DAB staining also showed less $O_{2}^{-}$ and H_2_O_2_ accumulation in the OE plants compared to the WT and EV plants (Figure 8M,N). There was no significant difference in SOD, CAT, and POD at 0 dpt (*p* > 0.05). The difference was that with the increase of time, the activity levels of three enzymes in OE plants were higher than those in the control group (WT, EV), and the ROS levels were lower (Figure 8D–F,J–L). These results indicated that the *N. tabacum* overexpressing *IlWRKY22* had a stronger ability to remove ROS and accumulate fewer harmful substances under salt and drought stresses than the control and could better cope with salt and drought stresses.

### 3.10. The IlWRKY22 on Expression Levels of Stress-Related Genes NtCAT, NtHAK1, NtPMA4, NtPOD, NtSOD, and NtSOS1 under Salt and Drought Stress

To further investigate the role of *IlWRKY22* in enhancing tolerance to salt and drought stress, we quantified the relative expression levels of stress-related genes. Our results revealed that *NtCAT*, *NtHAK1*, *NtPMA4*, *NtPOD*, *NtSOD*, and *NtSOS1* were significantly up-regulated under both salt and drought stress, regardless of the genetic background (Figure 9). We found that the relative expression levels of these genes were significantly up-regulated in all the tobacco plants at 7 and 14 dpt, but the increase of each gene in the OE-2, OE-3 and OE-5 lines was significantly higher than that of the WT and EV at both time points. However, OE lines showed the highest degree of increase at both 7 and 14 dpt compared to that of the control plants.

## 4. Discussion

Using our *I. laevigata* transcriptome data from the tepals, we identified a total of 68 WRKYs and classified them into three groups: 22 in Group I, 35 in Group II, and 11 in Group III. These three groups of WRKYs may participate in various biological processes or pathways based on previous functional studies in multiple plant species (Appendix A). The functional redundancy of these WRKYs indicates their key roles in responding to both external and internal stimuli [51]. It can also be reflected by the presence of redundant sequences in many plant species including *Xanthoceras sorbifolium* Bunge. [52] and *Solanum lycopersicum* L. [53], possibly due to duplication during plant evolution. Indeed, several sub-genome duplication events may have led to the formation of a large WRKY family in most modern eukaryotes [33].

*IlWRKY22* may be a multifaceted TF controlling multiple traits [13,18,19,54]. Thus, we selected it for functional studies. We focused on its role in flowering and found that overexpressing *IlWRKY22* delays flowering in *A. thaliana* and *N. tabacum*. This provides another line of evidence to support the key roles of WRKYs in controlling flowering time. For example, heterologous expression of the cotton *GbWRKY1* gene in *A. thaliana* promotes flowering by regulating the transcription of *SOC1* [12]. Similarly, an up-regulation of *BcWRKY22* in Chinese cabbage by low temperature promotes bolting and flowering via increasing *BcSOC1* [55] and metabolism of other players for flowering control including *SPL3*. A lower-level expression of *SPL3* in the transgenic plants may further lead to a down-regulation of *SOC1*, which ultimately causes a delay in the environment and initiates dynamic regulation of flowering [56].

Corresponding to the multiple environmental aspects affecting flowering, plants evolved multiple flowering transition pathways including the de-repression of *CDF* on *CO* and *FT* by the photoperiod pathway; the de-repression of *FLC* on *FT* in the vernalization, temperature, and autonomous pathways; as well as the de-repression of *SOC1* by degrading DELLA proteins in the GA pathway. To delineate the exact mechanisms in which *IlWRKY22* was involved, we quantified the expression of 10 genes in multiple pathways (Figure 5). Our data showed a significant up-regulation of *CO* and *GA20OX*, indicating that the primary networks downstream of *IlWRKY22* were the photoperiod pathway and the GA pathway [48]. This in turn resulted in altered metabolism of other players for flowering control including *SPL3*. A lower-level expression of *SPL3* in the transgenic plants may further lead to a down-regulation of *SOC1*, which ultimately causes a delay in flowering. Interestingly, *SPL3* is a major effector in the aging pathway as overexpressing or knocking-out dark-treated *AtWRKY22* led to accelerated and delayed senescence phenotypes, respectively, in *A. thaliana* [54]. Thus, it raises the possibility that *IlWRKY22* regulates flowering transition via the aging pathway and that *IlWRKY22* is the converging point of multiple pathways including flowering and senescence.

In addition to flowering control, *WRKY22* also plays a role in abiotic stress in plants. For example, *CsWRKY22* in *Citrus sinensis* regulates susceptibility to canker disease [57], *LiWRKY22* can promote heat resistance of *lily* [58]; *CmWRKY22* has a positive regulatory effect in response to drought stress [59]. Our study further revealed a role of *IlWRKY22* in responding to salt and drought stress [60]. Compared to controls, plants overexpressing *IlWRKY22* showed much-improved growth under both salt and drought stress (Figure 6). This is in line with the finding that *WRKY22* is significantly up-regulated by drought stress. The enhanced resistance to salt and drought stress can be attributed to up-regulation in photosynthesis (higher Chl content, Pn, and Fv/Fm compared to the control plants) and water retention capacity (lower Gs, Ci, and Tr, Figure 7). In this case, overexpressing *IlWRKY22* in *N. tabacum* still has higher photosynthetic capacity and water retention capacity compared with WT *N. tabacum*, as indicated by relatively higher Chl content and lower Gs. This is consistent was a previous report [61]. The lower Ci in transgenic plants can be explained by higher photosynthetic activities [39]. The ability of *IlWRKY22* in enhancing the resistance of *N. tabacum* to salt or drought stress and maintaining growth in stress environments are interconnected. Photosynthesis is an important indicator of the ability of plants to respond to abiotic stress [62]. Under high-salt and drought conditions, plants in general lose water due to high osmotic pressure, leading to stomatal closing, disrupted chloroplast structures, decreased activity of chlorophyll enzymes, and eventually inhibition of photosynthesis [63,64,65]. Manipulating the expression of protective genes and thus the regulation of gene networks may confer a better resistance. Our data suggested that the protective role of *IlWRKY22* involved multiple processes that can alleviate the adverse impact of salt and drought stress.

Although the exact molecular mechanisms by which *IlWRKY22* protects plants against salt and drought stress are not clear, one possible mechanism is via ROS regulation. This is evidenced by the observation that the transgenic plants overexpressing *IlWRKY22* accumulate less $O_{2}^{-}$, H_2_O_2_, and MDA compared to that of the control plants (Figure 8). The generation of ROS and secondary damage to the cell membrane including membrane lipid peroxidation under environmental challenges have been well documented in plants [66,67]. By lowering the level of oxidative stress, *IlWRKY22* may protect plants from excessive damage to the cellular membrane under stress. In addition, numerous studies have shown the key role of antioxidant enzymes including SOD, POD, and CAT in counterbalancing ROS in plants [66,68]. Consistent with these studies, we found that the enzyme activities of SOD, CAT, and POD were increased in *IlWRKY22*-overexpressing plants under stress (Figure 8). In summary, both an increase in antioxidant enzyme activity and a decrease in ROS accumulation led to a lower level of oxidative stress under salt and drought treatment in transgenic plants, which contributed to a better photosynthetic capacity and enhanced resistance under unfavorable conditions.

To further investigate the role of *IlWRKY22* under salt and natural drought stress, we examined the expression levels of several key genes associated with plant stress tolerance in transgenic plants overexpressing *IlWRKY22* following stress induction. Our results revealed a significant up-regulation of *NtCAT*, *NtHAK1*, *NtPMA4*, *NtPOD*, *NtSOD*, and *NtSOS1* in the transgenic *N. tabacum* plants compared to the control plants. This suggests that *IlWRKY22* may directly or indirectly regulate the expression of downstream stress response genes. Salt overly sensitive 1 (SOS1) plays a crucial role in facilitating the efflux of Na^+^ ions from cells, thereby protecting plants from salt-induced toxicity caused by excessive intracellular Na^+^ accumulation [69]. Plasma membrane H^+^-ATPase4 (PMA4) is involved in regulating cellular K^+^ uptake and maintaining appropriate K^+^ concentration [70]. Additionally, the high affinity kalium transporter1 (HAK1) helps balance intracellular Na^+^/K^+^ levels and prevents excessive intracellular Na^+^ content from causing cellular toxicity [71].

## 5. Conclusions

*IlWRKY22* has the dual function of flowering time control and regulation in salt and drought stress. Overexpressing *IlWRKY22* in *A. thaliana* and *N. tabacum* causes a delay in flowering via the effect of the expression of *CO*, *GA20OX*, *SPL3*, and *SOC1*. In addition, overexpression of *IlWRKY22* in *N. tabacum* enhances the tolerance to salt and drought treatment by positively regulating the antioxidant system and photosynthesis. Thus, *IlWRKY22* is a promising candidate for breeding *Iris*, as it has the potential to delay the flowering period of *Iris* and enhance the plant’s resistance to salt and drought.

## Figures and Tables

**Figure 1 plants-13-01191-f001:**
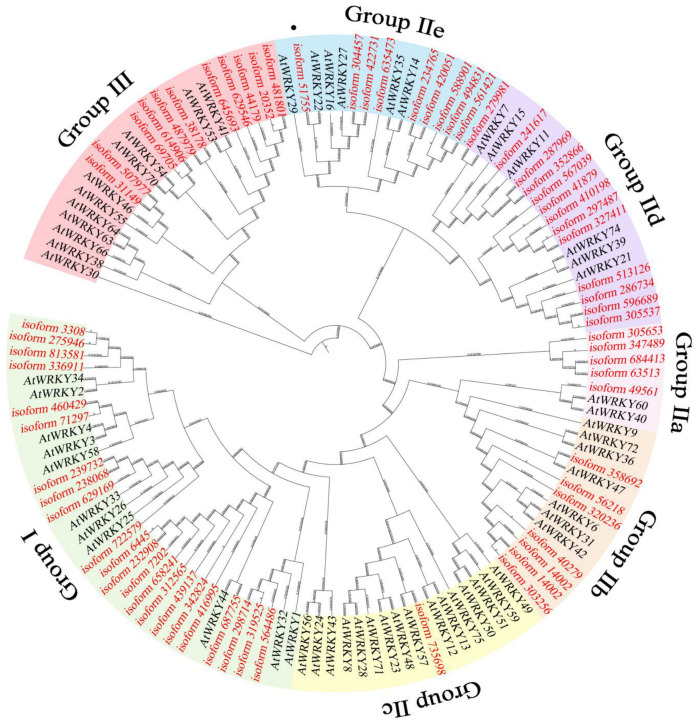
Phylogenetic analysis of *WRKYs* from *I. laevigata* (in red) and *A. thaliana* (in black). Black dots indicate *IlWRKY22* from *I. laevigata*.

**Figure 2 plants-13-01191-f002:**
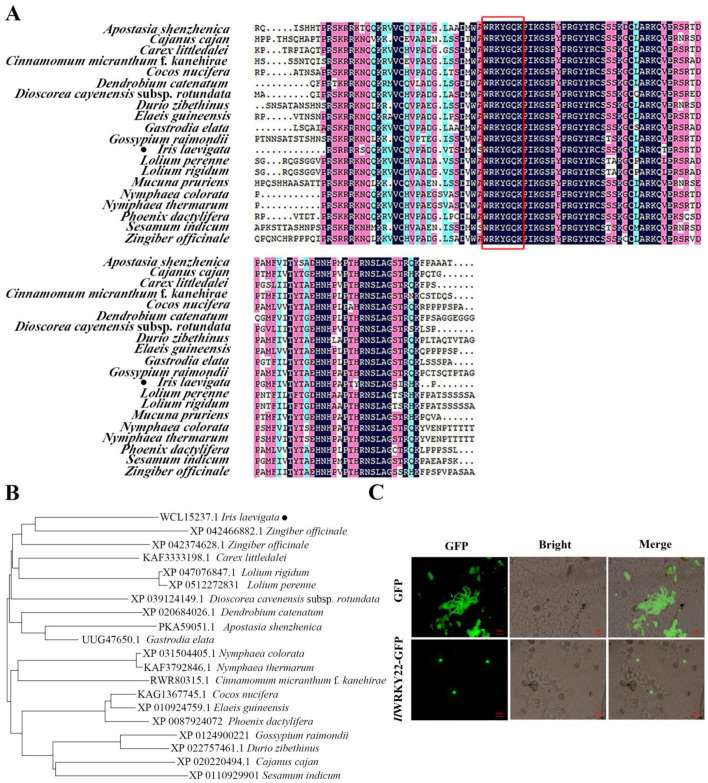
Sequence phylogenetic analysis and subcellular localization of *IlWRKY22*. (**A**) Protein sequence alignment of IlWRKY22 and that of other plant species (*Zingiber officinale*, *Zingiber officinale*, *Carex littledalei*, *Lolium rigidum*, *Lolium perenne*, *Dioscorea cavenensis* subsp. *rotundata*, *Dendrobium catenatum*, *Apostasia shenzhenica*, *Gastrodia elata*, *Nymphaea colorata*, *Nvmphaea thermarum*, *Cinnamomum micranthum* f. *kanehirae*, *Cocos nucifera*, *Elaeis guineensis*, *Phoenix dactvlifera*, *Gossypium raimondii*, *Durio zibethinus*, *Cajanus cajan*, and *Sesamum indicum*). The red box is the highly conserved WRKYGQK domain. Black dots indicate IlWRKY22 from *I. laevigata*. Among them, purple represents 100% homologous sequence, pink represents ≥75% homologous sequence, blue represents ≥50% homologous sequence, yellow represents ≥33% homologous sequence. (**B**) A phylogenetic tree of IlWRKY22. Black dots indicate IlWRKY22 from *I. laevigata*. (**C**) Subcellular localization of IlWRKY22 protein. From left to right: GFP, bright field and merge signals. Scale bar was 20 μm.

**Figure 3 plants-13-01191-f003:**
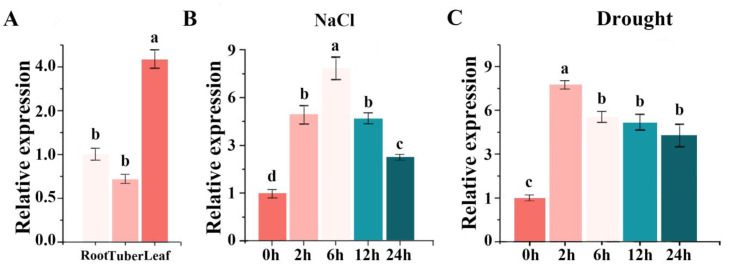
Expression analysis of *IlWRKY22*. (**A**) The expression of *IlWRKY22* was analyzed in the roots, tubers, and leaves of *I. laevigata.* (**B**) The expression of *IlWRKY22* was examined under NaCl stress at different time points. (**C**) The expression of *IlWRKY22* was evaluated in response to PEG-6000 at different time intervals. The bar chart in the figure displays the results, with different letters on top indicating significant differences between the groups (*p* < 0.05).

**Figure 4 plants-13-01191-f004:**
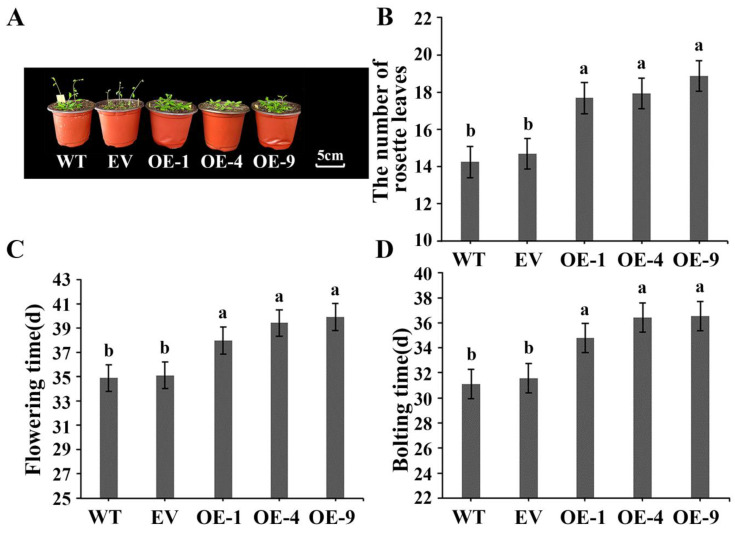
Phenotypes of transgenic *A. thaliana* overexpressing *IlWRKY22*. (**A**) Phenotypic observation of transgenic *A. thaliana* overexpressing *IlWRKY22*. Scale bar was 5 cm. (**B**) The number of rosette leaves at the time of flowering. (**C**) Flowering time (opening of the first flower). (**D**) Bolting time (inflorescence stalk of 1 cm). Bar graph data are Mean ± SD, marked with different lowercase letters to indicate significant differences (*p* < 0.05). WT is wild type plants. EV is empty vector plants. OE-1, OE-4, and OE-9 are overexpressed lines of *IlWRKY22*. All these experiments were performed in triplicate.

**Figure 5 plants-13-01191-f005:**
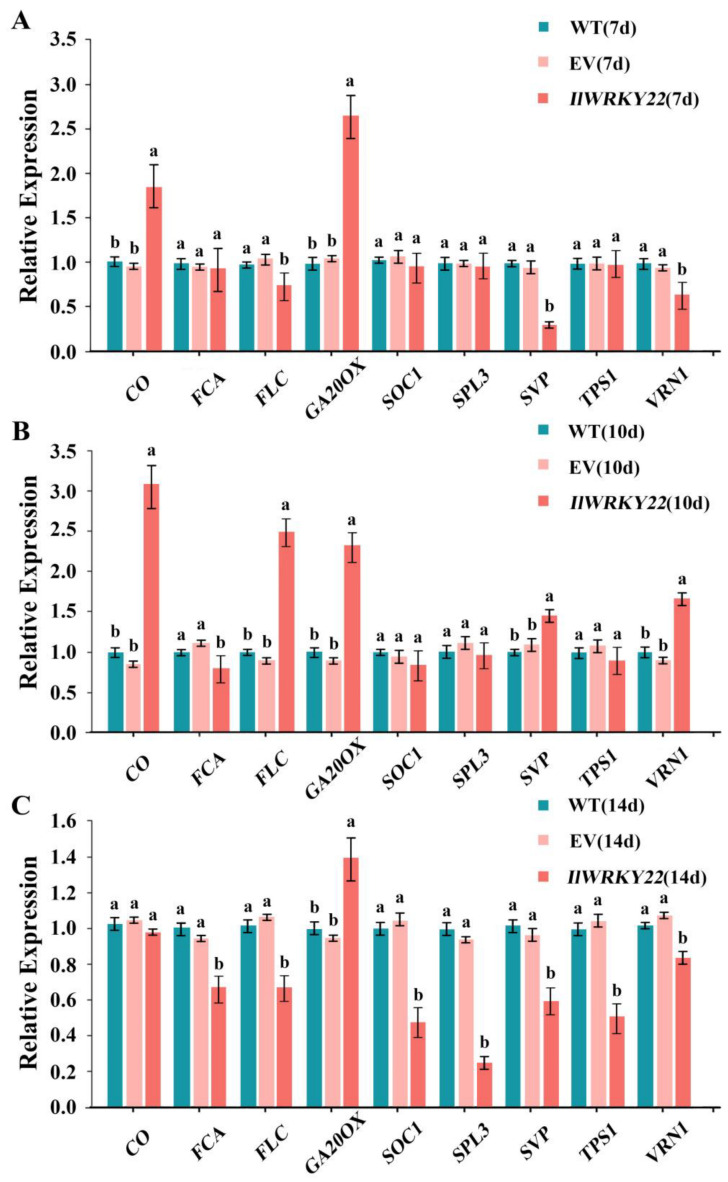
Expression of endogenous flowering time genes in transgenic *A. thaliana* overexpressing. (**A**) In 7-day-old seedlings. (**B**) In 10-day-old seedlings. (**C**) In 14-day-old seedlings. Bar graph data are Mean ± SD, marked with different lowercase letters to indicate significant differences (*p* < 0.05). All of these experiments were performed in triplicate.

**Figure 6 plants-13-01191-f006:**
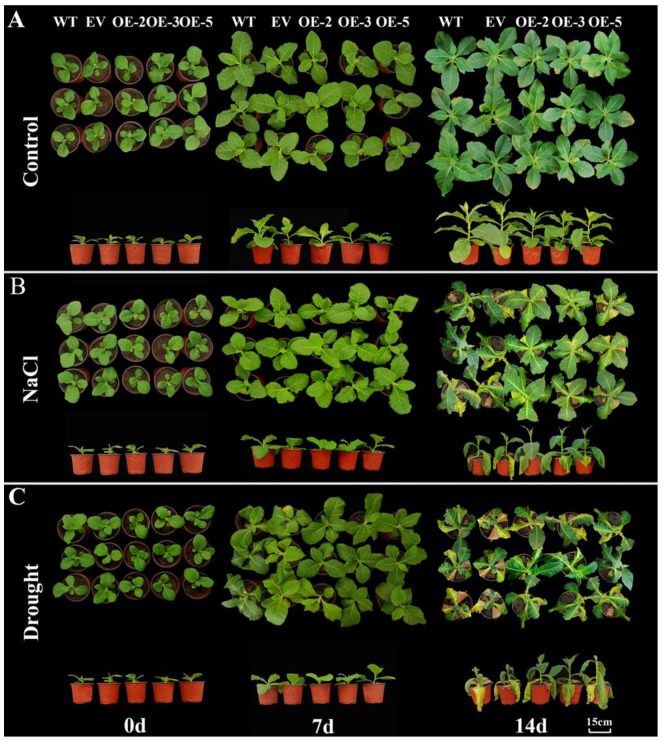
Overall plant phenotypes under salt and drought stress at different days post-treatment. (**A**) Top view and elevation of the growth state of five lines of *N. tabacum* at different in the normal growth state. (**B**) Top view and elevation of the growth status of five lines of *N. tabacum* at different days under NaCl stress. (**C**) Top view and elevation of the growth status of five lines of *N. tabacum* at different days under natural drought stress. The scale bar is 15 cm.

**Figure 7 plants-13-01191-f007:**
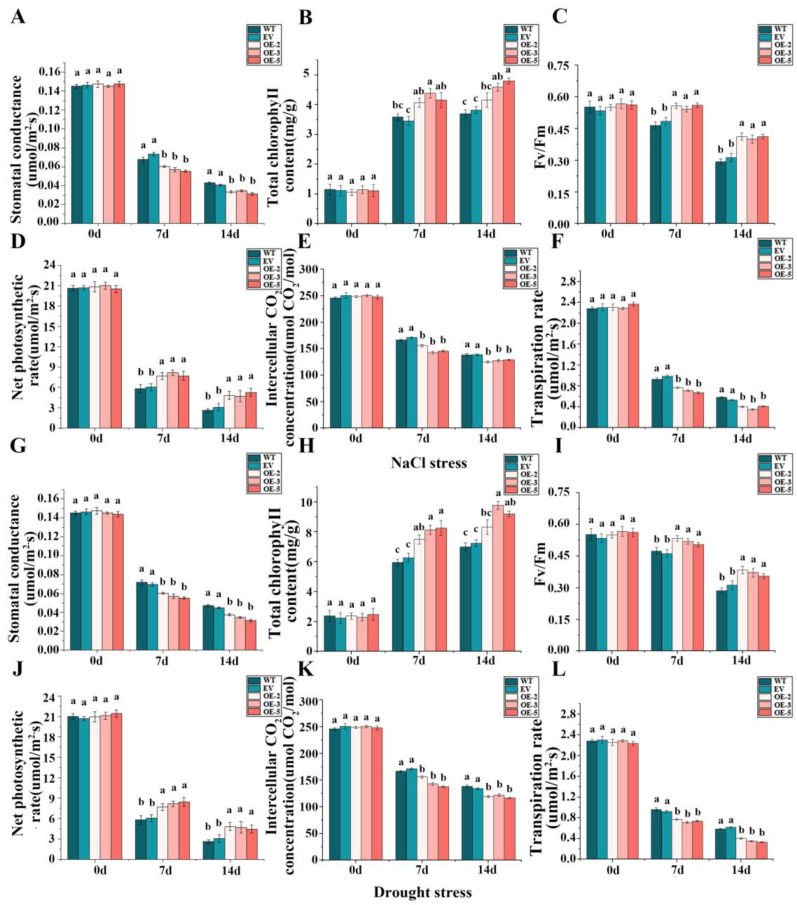
Impact of overexpressing *IlWRKY22* on photosynthetic and transpiration rate indexes under salt and drought stress. (**A**–**F**) Gs, Chl content, Fv/Fm, Pn, Ci, and Tr of *N. tabacum* five lines at different days under NaCl stress. (**G**–**L**) Gs, Chl content, Fv/Fm, Pn, Ci, and Tr of *N. tabacum* five lines at different days under natural drought. Different letters above the bars indicated significant differences (*p* < 0.05). WT is wild type plants. EV is empty vector plants. OE-2, OE-3, and OE-5 are overexpressed lines of *IlWRKY22*. All these experiments were performed in triplicate.

**Figure 8 plants-13-01191-f008:**
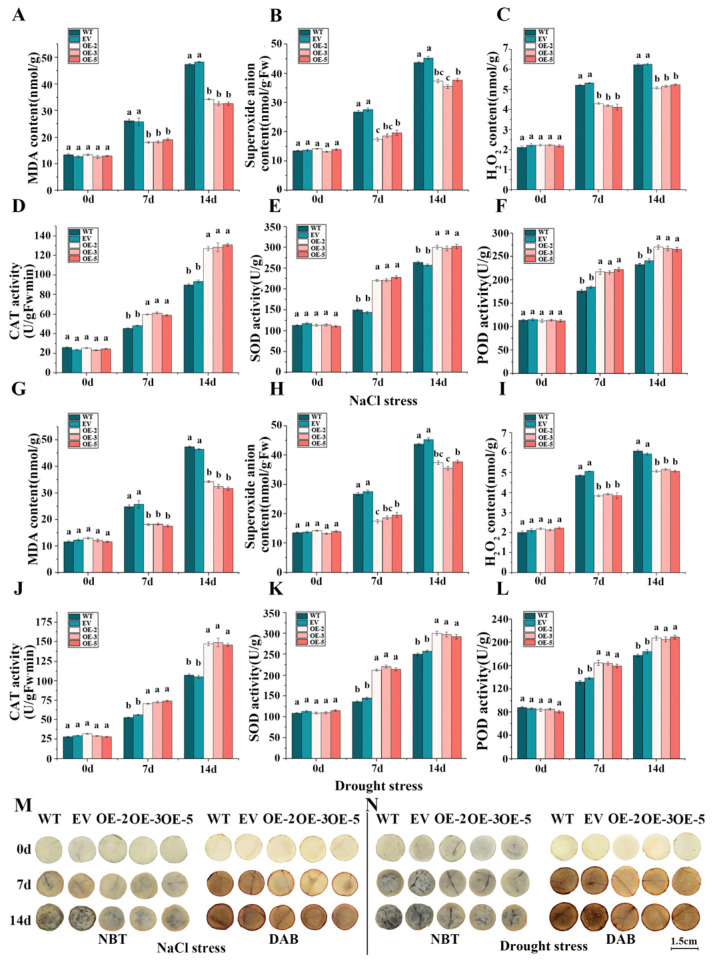
Involvement of ROS regulation. Analysis of physiological indexes of *N. tabacum* seedlings overexpressing *IlWRKY22* under salt stress and drought stress. (**A**–**F**) The contents of MDA, $O_{2}^{-}$, H_2_O_2_ and the activities of CAT, SOD, and POD in *N. tabacum* five lines leaves were measured at different days after NaCl stress. (**G**–**L**) The contents of MDA, $O_{2}^{-}$, H_2_O_2_ and the activities of CAT, SOD, and POD in *N. tabacum* five lines leaves were measured at different days of natural drought stress. Different letters on the bar chart represent significant differences (*p* < 0.05). (**M**) Plot of leaf disc staining using NBT and DAB in five lines at different days of NaCl stress. (**N**) Leaf disc staining plots using NBT and DAB in five lines at different days of natural drought stress. A deeper NBT staining indicates a higher accumulation of $O_{2}^{-}$, while a deeper DAB staining indicates a greater accumulation of H_2_O_2_. Consequently, increased levels of both $O_{2}^{-}$ and H_2_O_2_ accumulation led to more severe plant damage. The scale bar is 1.5 cm.

**Figure 9 plants-13-01191-f009:**
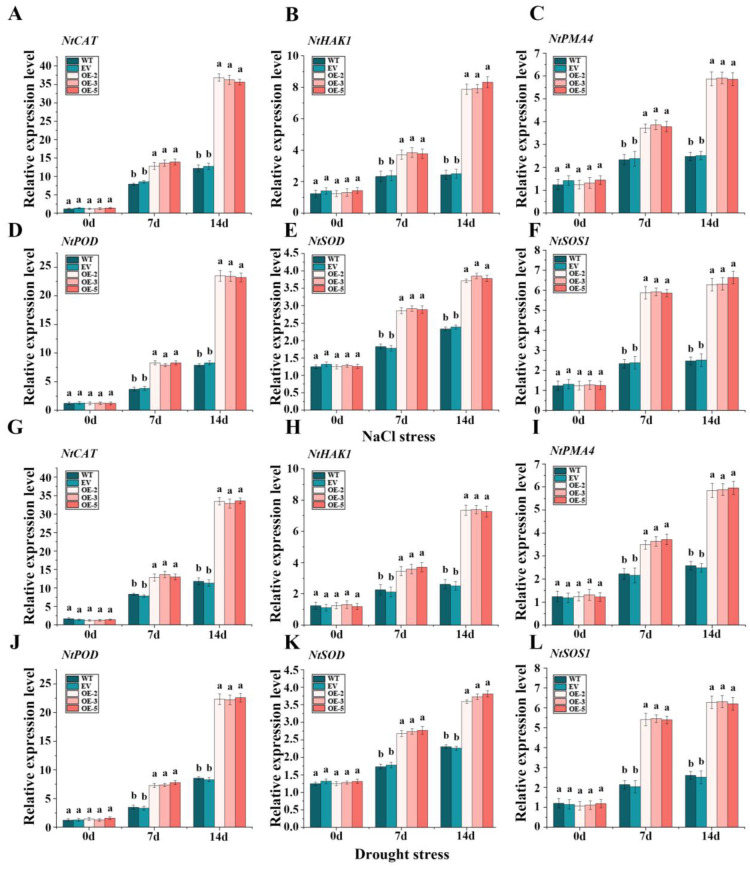
Expression of salt and drought stress-responsive genes in overexpressed *IlWRKY22 N. tabacum*. (**A**–**F**) The expression levels of *NtCAT*, *NtHAK1*, *NtPMA4*, *NtPOD*, *NtSOD*, and *NtSOS1* gene in five lines at different days under 300 mM NaCl. (**G**–**L**) The expression levels of *NtCAT*, *NtHAK1*, *NtPMA4*, *NtPOD*, *NtSOD*, and *NtSOS1* gene in five lines at different days under natural drought. Different letters above the bars indicated significant differences (*p* < 0.05).

## Data Availability

Data from this study are available from the corresponding author upon reasonable request.

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
