# Peer review of "WRKY22 Transcription Factor from Iris laevigata Regulates Flowering Time and Resistance to Salt and Drought"

_plants, 2024, doi:10.3390/plants13091191_

Round 1

Reviewer 1 Report

Comments and Suggestions for Authors

Please find enclosed my comments on the MS entitled : « WRKY22 Transcription Factor from Iris laevigata Regulates Flowering Time and Resistance to Salt and Drought »   by Fan et al.

WRKY TFs have been widely studied for their involvment in response to stress. More recently a function in flowering and development has been suggested. The aim of this work is to investigate the potential use of WRKY22 of Iris laevigata in breeding programs to modify both flowering time and response to stress.  The authors identified the sequences of WRKY genes in Iris laevigata. They found 68 putative members. Phylogenetic analysis show that the distribution among the various groups is different from Arabidopsis. They focus on ilWRKY22 to study its function on flowering time and response to stress as it is described as a multifaceted TF. After showing that the protein was nulear localized in tobacco transient assay, they show that this gene was induced in response to salt and PEG stress in Iris laevigata. They generated many Arabidopsis and tobacco lines overexpressing ilWRKY22. They selected 3 lines of each to further perform analysis. They showed that in Arabidopsis the OE of ilWRKY22 delayed bolting and flowering time. This was less obvious  in tobacco. In Arabidopsis this phenomenom  was linked to changes in the expression level of key genes in the flowering pathway. They showed that in tobacco the OE of WRKY22 enhanced resistance to salt and natural drought stress. The authors performed on theses lines the measurments of many physiological parameters, gene expression of stress related gene, production of ROS, anti-oxydant enzymatic activities etc... They showed that this enhanced tolerance is due to a positive regulation of the antioxydant system and photosynthesis. All together they conclude that  ilWRKY22 is a good target gene for breeding programs in Iris laevigata.

Althought I did not see mention whether ilWRKW22 would have an impact on the duration of the flowering time for this ornementale plant,   I beleive there is a huge amount of work perform with the transgenic lines. Experiments are performed on 3 independant lines each time and measurments are done at 3 times points. Each physiological assay is  complemented with many molecular evidences. I think the work is robusted. However Bibliography is not always apropriate. I beleive the results need to be better described. Some mistakes are also written and some duplications of sentences are found. Also too many graphs with nearly no explanation provided in the result section.

Introduction :

Lines 71 : can we say TF interact with substrat ?

Lines 75-77 : The different groups of WRKY should be explained.

Lines 80-112 : please check references cited. Appropriate ? check all.

Material and method :

The statisical analysis is not clear.

Please describe better Gene Sequence Identification and Phylogenetic Analysis

Result :

Figure 2 : would be nice to see different cells expressing ilWRKY22-GFP

Figure 3 legend : in which tissu is qPCR performed in response  to stress ? I guess leaves. How many plants tested ?

Figure Sup1 : to study expression of the transgene, are the primers specific of ilWRKY22 or could they measured as well At WRKY22 ?

Figure 5 and Result section 3.6 : At  day 7, I see only 2 genes up regulated (and not many as said in the result section) and at day 10, more. Please check again the data.

Also : twice repeat « Notably, SPL3 showed 80% down-regulation in the OE plants. This indicated that the OE 343 plants were still at the early stage of flowering transition on day 14 »

Why all the genes tested are downregulated at day 14 ?

The authors did not comment whether the duration of the flowering period was changed as well.

Figure 6 and result section 3.7 : it is not clear to me why the authors did not check resistance to drought and salt on the OE lines in At.

Lines 359 : Each strain was replicated 15 times. What does it mean ?

This section is not well explained. Please reword.

Figure 7 section 3.8 : 12 graphs explained in 10 lanes. I guess some parts can be put in supplemental or please detailed a bit more. Stomatal ( and not stonmatal) conductance is not mentionned. Please explain what are the different parameters studied.

Figure 8 section 3.9 : same comment. Too many bar charts with no comment.

Figure 9 section 3.10 : same comment.

Discussion :

 « Thus, it raises the possibility that  IlWRKY22 regulates flowering transition via the aging pathway and that IlWRKY22 is the  converging point of multiple pathways including flowering and senescence 

Comments on the Quality of English Language

Author Response

Thank you very much for taking the time to review this manuscript. Please find the detailed responses below and the corresponding revisions in track changes in the re-submitted files.

Reviewer 2 Report

Comments and Suggestions for Authors

Fan et al., present an interesting research on the role of WRKY22 transcription factor in Iris laevigata in flowering time as well as in abiotic stress particularly Salt and drought stress

The study is an important advancement in the stress biology field specially with regards to ornamental plant such as Iris that poses serious issues when it comes to its utility as ornamental plants particularly due to its short flowering time. This study can open up the avenue for generating novel cultivars with improved flowering time and enhanced stress resistance to salt and drought. 

However, I have few questions for the authors and would appreciate if the authors can clarify: 

Material and methods: 

  1. I think the methods section should be little more details. Authors have mentioned “as described before” in some instances which is not suitable. It will be inconvenient to the readers to go back looking for all the protocols. The MS would benefit significantly from revising the material and method section

  2. The authors have used several abbreviations in the method section without mentioned at first mention at several places. Please correct this. 

Results:

  1. Page 6 (Line 260-264): instead of quantity in the bracket, you can write “n”

  2. Figure 3 Legend: It is very difficult to follow what parts of the plants were used in the qPCR analysis. This information should be clearly stated in the legend. 

  3. Page 9 (Line 316-317): Authors have alluded to an increase in the flowering time and number of rosette leaves in the OE lines. I am wondering how much of this increase is significant for the plants especially considering its utility as an ornamental plant.

  4. Figure 4 Legend: Please describe the abbreviation in the legend. 

  5. Page 10 (Line 347-349): This line seems to be repetitive. 

  6. I have some doubts about the results in Figure 5. It is not clear why the empty vector and WT showed differences in the expression of some of the genes , in some cases little higher than the WT. Is it expected, and if yes, can the authors provide a rationale behind it. 

  7. Figure 7 Legends: Please provide the full form of all Abbreviations. 

  8. What I find missing is a discussion on a possible link between two completely opposite functions of WRKY TFs in flowering time and stress response. Do authors have ideas on that and if they can relate that observation to some previous studies?

Comments on the Quality of English Language

Minor Editing of English Language Required at some places.

Author Response

(The authors gave the same response as above.)

Reviewer 3 Report

Comments and Suggestions for Authors

The manuscript “WRKY22 transcription factor from Iris laevigata regulates flowering time and resistance to salt and drought” by Fan et al is an interesting study related to the function of the transcription factor WRKY22 on plant development and plant responses to stress. 

Minor revision

There are some typo mistakes the manuscript must be carefully revised.

For example tabacum is written with capital letter in 2.8 and 2.9 sentences.

Figure 8, droughe must be changed to drought

Methods: There is no information of the tools used for gene sequences identification

Major revision:

Statistical analyses.  There is no information of the number of plants and biological replicates used for 2.8 and 2.11 sections.  The description of the experimental design must be improved

In the section “Stalistical Analyses” there is no information about the statistical tests that were used for the different experiments. This information has to be included in this section.

Subcellular Localization.  It is important to used a nucleus marker to determine nuclear localization of the protein

Author Response

(The authors gave the same response as above.)

Round 2

Reviewer 3 Report

Comments and Suggestions for Authors

The authors have considered all the suggestions and comments to improve the manuscript. The manuscript can be accepted in the present form.